

# A survey of RNA secondary structural propensity encoded within human herpesvirus genomes: global comparisons and local motifs

Ryan J. Andrews, Collin A. O'Leary and Walter N. Moss

The Roy J. Carver Department of Biochemistry, Biophysics and Molecular Biology, Iowa State University, Ames, IA, United States of America

## ABSTRACT

There are nine herpesviruses known to infect humans, of which Epstein–Barr virus (EBV) is the most widely distributed (>90% of adults infected). This ubiquitous virus is implicated in a variety of cancers and autoimmune diseases. Previous analyses of the EBV genome revealed numerous regions with evidence of generating unusually stable and conserved RNA secondary structures and led to the discovery of a novel class of EBV non-coding (nc)RNAs: the stable intronic sequence (sis)RNAs. To gain a better understanding of the roles of RNA structure in EBV biology and pathogenicity, we revisit EBV using recently developed tools for genome-wide motif discovery and RNA structural characterization. This corroborated previous results and revealed novel motifs with potential functionality; one of which has been experimentally validated. Additionally, since many herpesviruses increasingly rival the seroprevalence of EBV (VZV, HHV-6 and HHV-7 being the most notable), analyses were expanded to include all sequenced human Herpesvirus RefSeq genomes, allowing for genomic comparisons. In total 10 genomes were analyzed, for EBV (types 1 and 2), HCMV, HHV-6A, HHV-6B, HHV-7, HSV-1, HSV-2, KSHV, and VZV. All resulting data were archived in the RNAStructuromeDB (https://structurome.bb.iastate.edu/herpesvirus) to make them available to a wide array of researchers.

## INTRODUCTION

The herpesviruses are an ancient family of double-stranded DNA viruses (Order *Herpesvirales*, Family *Herpesviridae*). The three major subfamilies (clades) of mammalian herpesviruses—α (*Alphaherpesvirinae*), β (*Betaherpesvirinae*), and γ (*Gammaherpesvirinae*)—appeared ∼180 to 220 mya (*McGeoch et al., 1995*). Nine herpesviruses are currently known to infect humans: Epstein–Barr virus (EBV), human cytomegalovirus (HCMV), human herpesvirus 6A and 6B (HHV-6A and HHV-6B), human herpesvirus 7 (HHV-7), herpes simplex viruses 1 and 2 (HSV-1 and HSV-2), varicella-zoster virus (VZV), and Kaposi's sarcoma-associated herpes virus (KSHV). Of these, EBV is most prevalent with >90% of adults being infected (*Kieff & Rickinson, 2001*). EBV has a large (∼172 kb) double-stranded DNA genome that recapitulates many of the

Corresponding author
Walter N. Moss, wmoss@iastate.edu

biological processes important to the host genome (*Young & Rickinson, 2004*) including periods of pervasive transcription (*O'Grady et al., 2014*). After spreading through the host by lytic replication, EBV establishes latency in B cells via several distinct patterns of coding and noncoding gene expression. Both lytic and latent replication are associated with various *cancers* (e.g., lymphomas and carcinomas) (*Elgui de Oliveira, Muller-Coan & Pagano, 2016*) and *autoimmune diseases* (e.g., multiple sclerosis and Lupus) (*Draborg, Duus & Houen, 2013*); however, molecular mechanisms connecting EBV infection to disease remain elusive—making EBV an important target of study. One route for the regulation of viral/host biology and for mediating host-virus interactions is RNA secondary structure. Many key biological processes of both hosts and their pathogens are affected by RNA structure: e.g., RNA transcription, processing, stability, and expression are all affected by RNA folding (*Andrews & Moss, 2019*; *Bevilacqua et al., 2016*).

Although several functional RNA structural motifs have been identified in EBV (*Moss et al., 2014*), knowledge of the roles of RNA and RNA folding in EBV infection and disease still require additional attention. The first viral noncoding (nc)RNAs, discovered in 1981, were the E̲BV e̲ncoded R̲NAs (EBERs −1 and −2) (*Lerner et al., 1981*). Both EBERs are highly-expressed in infected cells (dwarfing endogenous ncRNA expression) and possess extensive secondary structure that mediates their interactions with host and viral biomolecules to promote infection (*Lee et al., 2015*; *Lee, Pimienta & Steitz, 2012*; *Lee et al., 2016*; *Moss et al., 2014*). Over 20 years later, additional viral ncRNAs were discovered: the BART and BHRF EBV-encoded m̲icro (mi)RNAs in 2004 (*Pfeffer et al., 2004*) and the EBV v̲iral s̲mall n̲ucleo̲lar (v-snoRNA)1 in 2009 (*Hutzinger et al., 2009*).

A genome-wide bioinformatics and experimental analysis was previously performed in EBV to scan for functional RNA motifs (*Moss & Steitz, 2013*). In the previous analysis, the ncRNA discovery program RNAz (*Gruber et al., 2010*) was used to analyze sliding windows of aligned EBV genome sequences (plus one closely related lymphocryptovirus from rhesus monkeys: Macacine herpesvirus 4) to predict windows that have unusual thermodynamic stability and conservation of RNA secondary structure—both indicators of functionality (*Clote et al., 2005*). Approximately 30% of the (∼170 kbp) genome was predicted to potentially encode functional RNA structure. In addition to known structures (EBERs, v-snoRNA1, and pre-miRNA hairpins), many novel structural elements were predicted. This analysis discovered a new class of EBV ncRNAs, the s̲table i̲ntronic s̲equence (sis)RNAs, which consist of abundant, highly structured intronic-sequence RNAs (derived from the EBV W repeats). The sisRNAs consist of a complex mix of excised introns and, more recently discovered, alternatively spliced long (l)ncRNAs (up to ∼22 kb) (*Cao et al., 2015b*; *Tompkins, Valverde & Moss, 2018*). The sisRNAs interact with multiple host regulatory proteins (*Tompkins, Valverde & Moss, 2018*) and are essential to EBV's ability to transform human B cells (*Bridges et al., 2019*). Very long (∼500 nt) hairpins were modeled in the latent origin of replication (oriP), which were later shown to occur in bidirectional lytic lncRNAs that are hyper A-to-I edited and promote lytic replication (*Cao et al., 2015a*). Elsewhere, a structural motif partially spans two exons and a whole intron in the EBV LMP2 gene; structure in this motif mediates interactions with splicing regulatory proteins and,

surprisingly, nuclear actin to affect LMP2 splicing (a novel role for actin) (*Kumarasinghe & Moss, 2019*).

Although this analysis of EBV led to interesting discoveries, there were several limitations to this previous approach: (1) motifs that were not conserved (e.g., motifs unique to EBV) were potentially missed; (2) multiple alternative models were predicted for the same nt in different analysis windows; and (3) subjectively selected regions, comprised of overlapping windows with favorable metrics, were selected for generating final models. To address this, in this current study we used a new computational pipeline for RNA secondary structure analysis, `ScanFold`, which addresses these limitations and has been successfully applied to the Zika and HIV-1 genomes (*Andrews, Roche & Moss, 2018*) as well as the human MYC mRNA (*O'Leary et al., 2019*). The `ScanFold` pipeline divides RNA structural motif discovery into two parts; the first step is accomplished using `ScanFold-Scan`, where individual sequences are analyzed using a sliding window approach. This generates overlapping windows covering each nt of the sequence. For each window `ScanFold-Scan` calculates 4 folding metrics: (1) The m̲inimum f̲ree e̲nergy (MFE) $\Delta G°$, which predicts optimal thermodynamic stability. (2) The $\Delta G°$ z-score, which measures the stability of native vs. random sequence. (3) A *p*-value for the $\Delta G°$ z-score, which is a quality control metric that calculates the fraction of randomized sequences with greater stability than native: *p*-values closer to 0 indicate a high confidence z-score. (4) The ED (e̲nsemble d̲iversity), which indicates the structural diversity of Boltzmann-weighted conformations calculated from a partition function. The second step, carried out by the program `ScanFold-Fold`, builds secondary structural models from base pairs likely to be functionally significant. `ScanFold-Fold` first compiles and averages `ScanFold-Scan` metrics for every base pairing arrangement (for every nucleotide) predicted across the sequence. The key metric used for selecting the most favorable bp for a nt is the $\Delta G°$ z-score$_{avg}$, which highlights bp that are *consistently* found in low z-score windows. This process was able to detect (and model with accuracy) all known structured motifs in the HIV-1 and ZIKV genomes (*Andrews, Roche & Moss, 2018*). Further analysis revealed the entirety of `ScanFold-Fold` results for HIV, ZIKV, and Hepatitis C genomes were consistent with experimentally derived models—where the models from regions with the most negative z-score$_{avg}$ values matched experimentally derived predictions and models in positive z-score$_{avg}$ regions correlated to unstructured regions (*Andrews, Baber & Moss, 2019*). These results showed that `ScanFold` can serve not only as a motif discovery pipeline (*Andrews, Roche & Moss, 2018*), but as a tool to generally characterize RNA structural landscapes (*Andrews, Baber & Moss, 2019*).

In this current study we revisit EBV using this new approach, finding evidence for additional functional elements in EBV including one that was experimentally characterized. Significantly, in this new study, focus is placed on the more prevalent EBV type 1 RefSeq genome (*Smatti et al., 2018*) (rather than the type 2 genome previously analyzed); this current study, however, also includes data for the type 2 genome. Additionally, all sequenced human herpesvirus RefSeq genomes have been analyzed to allow for global comparisons between genomes and to facilitate the identification of specific viral RNA motifs for further structure/function analyses. All results for EBV and other human herpesviruses are archived
on the RNAStructuromeDB a public resource archiving RNA secondary structural data for humans and our pathogens (*Andrews, Baber & Moss, 2017*).

## METHODS

Human herpesvirus genome sequences were acquired from the NCBI RefSeq database (*O'Leary et al., 2016*): EBV-1 (NC_007605.1), EBV-2 (NC_009334.1), HCMV (NC_006273.2), HHV-6A (NC_001664.4), HHV-6B (NC_000898.1), HHV-7 (NC_001716.2), HSV-1 (NC_001806.2), HSV-2 (NC_001798.2), KSHV (NC_009333.1), VZV (NC_001348.1). Each sequence was analyzed using the `ScanFold` pipeline. First, genome sequences were scanned using a sliding analysis window of 150 nt with a single nt step size using `ScanFold-Scan`. In each analysis window, the minimum Gibbs free energy of folding ($\Delta G$) and its associated secondary structure were predicted, assuming the genome sequence was transcribed into RNA. A $\Delta G$ z-score is calculated by comparing the native sequence to the average $\Delta G$ of 50 randomized sequences (shuffling nucleotides) according to the following equation: z-score $= (\Delta G_{native} - \Delta G_{random,average}) / \sigma$; here, $\sigma$ is the standard deviation of all calculated $\Delta G$s. Negative z-scores indicate the number of standard deviations more stable the native sequence is vs. random. A *p*-value is calculated, which is a quality control metric for the z-score that measures the fraction of random sequences more stable than native (ideally, this will be close to zero). The ensemble diversity is calculated from a partition function, which measures the diversity of folds in the possible folding ensemble. Here, low numbers indicate that there is a single dominant conformation, while higher numbers indicate alternative conformations or a lack of stable structure. From the partition function a centroid structure is also calculated, this is the secondary structure model that has the least difference with all other Boltzmann weighted conformations in the ensemble—thus, it can be considered as a structure that represents the ensemble fold. The parameters used for window/step size and randomizations were previously optimized (*Andrews, Roche & Moss, 2018*).

The output of `ScanFold-Scan` were then analyzed using the program `ScanFold-Fold`. Here, unique motifs are generated by creating consensus secondary structures across all overlapping scanning windows, where base pairs are weighted by the z-score of the window in which they occur. Motifs are thus generated from the base pairs that most contribute to the unusual structural stability of a sequence. Presumably, the evolved order of these sequences creates the particular stability of these pairs, indicating their potential for function (e.g., evolution working to stabilize such structures). A key component of `ScanFold-Fold` is the decision-making process used to define base pairs when multiple, unusually stable, pairs can be predicted for a residue; here, the algorithm selects the pairing partner that most consistently contributed to low z-score windows throughout the scan (see Methods of *Andrews, Roche & Moss, 2018*). For regions of interest, these unusually stable base pairs were used as constraints for generating global secondary structural models. This was done to "fill in" base pairs that were "missing" due to the generation of the consensus structure (e.g., where multiple, equally stable, pairs were averaged out) and to deduce potential long-range interactions, which spanned regions larger than the (150 nt)

windows used. To do this, the program `RNAfold` (*Lorenz et al., 2011*) was used to predict the minimum free energy secondary structure possible, given the constraint of forming all `ScanFold-Fold` with average z-score bp < −1 or −2 (described as the "Global Refold" option in *Andrews, Baber & Moss, 2019*).

To test predicted models for these select regions, a comparative sequence/structure analysis was performed. The EBV-1 sequence was used in a BLASTn analysis to deduce homologous sequence. These were aligned using the program `MAFFT` (*Katoh, Asimenos & Toh, 2009*; *Katoh & Frith, 2012*; *Katoh et al., 2005*; *Katoh et al., 2002*; *Katoh, Rozewicki & Yamada, 2017*; *Katoh & Standley, 2013*; *Katoh & Standley, 2016*; *Katoh & Toh, 2008a*; *Katoh & Toh, 2008b*; *Katoh & Toh, 2010*; *Kuraku et al., 2013*; *Nakamura et al., 2018*; *Yamada, Tomii & Katoh, 2016*) implementing the G-INS-i strategy, which is optimized for sequences with global homology. The `ScanFold-Fold` constrained secondary structure model for EBV-1 was mapped to this structure and conservation of base pairing was assessed as the percentage of canonical pairing (GC/CG, AU/UA, or GU/UG base pairs) observed across aligned positions. Mutations were tracked across model base pairs to identify structure-preserving mutations that were potentially consistent (single point mutations) or compensatory (double point mutations) with respect to the structure model.

To experimentally test RNA structural motifs for their effects on post-transcriptional gene regulation, select sequences were inserted into the pIS2 vector downstream of its R̲enilla L̲uciferase (RL) gene (within the 3′ UTR). All conditions were tested in biological triplicate as described in (*O'Leary et al., 2019*). Briefly, RL constructs, along with F̲iref̲ly (FF) Luciferase were transfected into HeLa cells and incubated at 37 °C and 5% $CO_2$ while maintained in DMEM supplemented with 10% FBS, penicillin, streptomycin, and L-glutamine. After 24 h, transfected cells were split between a 24 well plate (for qPCR) and a 96 well plate (for luciferase assays) and incubated an additional 24 h before analysis. Dual luciferase assays were designed and performed following recommendations of (*Van Etten, Schagat & Goldstrohm, 2013*): cells were lysed, and luciferase activity was measured using Promega's Dual Luciferase Reagent Assay kit with a collection time of 10 s. R̲elative r̲esponse r̲atios (RRR), calculated as the ratio of RL to FF r̲elative l̲ight u̲nits (RLUs), were calculated for each sample (Table S2).

The RT-qPCR analysis was carried out as described in (*O'Leary et al., 2019*): briefly, whole cell RNA was extracted in TRIzol followed by a Dnase I treatment (NEB) for 2 h at 37 °C. RNA was further purified with Zymo's RNA Clean and Concentrator kit before reverse transcription which was carried out using 1 μg of purified RNA, random hexamers, and Superscript III (ThermoFisher). Transcript abundance was then measured via qPCR and analyzed using the ΔΔCt method, where the relative abundance of RL was calculated using FF as the reference gene. The results of both RT-qPCR and dual luciferase assays can be seen in Table S2.

## RESULTS

### Global results

Average MFE ΔG values across each genome ranged from −24.8 kcal/mol (HHV-7) to −64.1 kcal/mol (HSV-2; Table 1). This large difference in predicted RNA folding stability,

**Table 1 ScanFold metrics of the human herpesviruses.** ED (ensemble diversity) is a measure of conformational diversity of RNA. Average Genome ZS (z-score) gives the ΔG z-score averages across all windows spanning each genome. ZS < −1 and < −2 give the percentage of prediction windows with G z-score below each cutoff. Total base pairs gives the total number of stable pairs predicted in each genome (counting both strands and all potential base pairs per nucleotide) and the bp ZS < −1 and < −2 give the percentages of bp below each z-score threshold. Motifs reports the number of individual discrete RNA structures (defined as having a single base helix) containing at least one bp with ZS < −2.

| Virus | GC% | ΔG (kcal/mol) | ED | Average ZS | Windows | ZS <-1 | ZS <-2 | Total bp | bp ZS <-1 | bp ZS <-2 | Motifs |
|---|---|---|---|---|---|---|---|---|---|---|---|
| EBV-1 | 59.4% | −50.1 | **32.2** | −0.63 | 343,348 | **35.1%** | **13.2%** | 835,324 | **8.8%** | 2.5% | 858 |
| EBV-2 | 59.6% | −50.3 | 32.3 | −0.62 | 345,230 | 34.4% | 12.7% | 850,827 | 8.4% | **2.6%** | **870** |
| KSHV | 53.7% | −42.6 | 32.6 | −0.31 | 275,640 | 25.0% | 7.9% | 628,314 | 6.4% | 1.5% | 381 |
| HSV-1 | 68.3% | −60.8 | 32.7 | −0.26 | 304,146 | 24.4% | 8.1% | 856,374 | 5.4% | 1.4% | 436 |
| HHV-7 | 36.2% | −24.8 | 33.7 | −0.26 | 305,862 | 22.8% | 7.6% | 536,275 | 6.8% | 1.6% | 320 |
| HSV-2 | **70.4%** | **−64.1** | 32.7 | −0.20 | 309,052 | 22.8% | 7.5% | 905,657 | 5.0% | 1.2% | 648 |
| HCMV | 57.4% | −46.8 | 32.9 | −0.19 | **470,994** | 22.9% | 6.9% | **1,143,973** | 2.9% | 0.6% | 562 |
| VZV | 46.0% | −34.5 | 32.9 | −0.13 | 249,470 | 20.0% | 5.1% | 498,533 | 5.7% | 1.0% | 226 |
| HHV-6A | 42.4% | −29.7 | 32.7 | −0.12 | 318,458 | 19.6% | 5.4% | 597,872 | 5.9% | 1.2% | 274 |
| HHV-6B | 42.8% | −29.7 | 32.9 | −0.09 | 323,930 | 19.0% | 4.7% | 607,730 | 5.8% | 0.8% | 268 |

as one would expect, correlates with the GC%, which ranged from 36.2% to 70.4%. The average z-score and underline{ensemble} underline{diversity} (ED), however, do not follow trends for ΔG or GC%. The z-score quantifies the greater-than-random folding stability of an RNA sequence and is primarily dependent on the sequence *order* and not its composition. Likewise, the ED value indicates the diversity of potential structural conformations in an RNA's folding ensemble, which also appears to be an evolved property of ordered/functional RNA sequences (*Moss, 2018a*). The average z-score ranged from -0.09 (HHV-6B) to −0.63 for EBV-1, while the average ED ranged from 33.7 (HHV-7) to 32.3 (EBV-1). Indeed, all folding metrics suggested functional structured motifs were most prevalent in the EBV type 1 genome.

Unsurprisingly, the predictions for EBV-1 and EBV-2 were almost identical (Table 1), as the two genomes are only ∼5% different in sequence and the majority of this difference clusters within one gene region (EBNA-2) (*Tzellos & Farrell, 2012*). The percentage of the EBV genome spanned by windows with z-scores <-1 (one standard deviation more stable than random) was predicted to be 35.1% and 34.4% for types 1 and 2, respectively. This is consistent with previous predictions using RNAz, which predicted ∼30% of the genome to fall within low z-score windows (*Moss & Steitz, 2013*). These values are stronger than any other human herpesvirus genome: e.g., the next highest percentage of windows with z-score <-1 were for KSHV (25%). This also held true for windows with z-scores <-2 (two standard deviations more stable than random), where EBV-1 and EBV-2 had 13.2% and 12.7% of their windows below this stricter cutoff: the next highest percentage was for HSV-1 (8.1%). The virus with the smallest fraction of its genome spanned by low z-score windows was HHV-6B (19.0% and 4.7% for the -1 and -2 z-score cutoffs, respectively).

In the prediction of the MFE ΔG for each analysis window, a model secondary structure is also generated. Across each genome this resulted in many predicted base pairs (e.g., the HCMV genome has 1,134,919 predicted base pairs; Table 1), where a specific nucleotide can

be paired differently across several overlapping windows. As a result, *many potential base pairing partners may be predicted per nucleotide*, resulting in the total number of predicted base pairs to often exceed genome length; a confounding factor which has proved to be a challenge in RNA 2D structure modeling (see Introduction). The `ScanFold-Fold` algorithm confronts this challenge by predicting the single most likely orientation for each nucleotide based on its contributions to low z-score windows (indicating ordered stability and, potentially, function). The viruses that had the greatest percentages of low z-score base pairs were EBV-1 and EBV-2: EBV-1 had 8.8% of its base pairs predicted with average z-score <-1 and EBV-2 had 2.6% of its base pairs predicted with average z-score <-2 (Table 1). Additionally, `ScanFold-Fold` extracts each discrete structural motif (i.e., single hairpins or multi-branched stem loops) containing at least one base pair with an average z-score <-2. This results in a list of motifs for each genome, where again, EBV-1 and EBV-2 had the greatest number of motifs with 858 and 870 respectively.

In summary, all predictions indicate a particular importance for functional RNA secondary structures encoded within the EBV genome vs. other herpesviruses. The `ScanFold-Fold` results also give us a means of generating motifs of interest for further analysis.

## Known structural motifs in EBV

The `ScanFold` analysis was able to deduce the location and, in many cases, the secondary structure of known RNA structural motifs in EBV-1. All known EBV-1 miRNA sequences fell within low z-score regions and were contained within hairpins formed by low z-score base pairs. For example, Fig. 1 summarizes `ScanFold` results for the EBV-1 BART miRNA gene cluster. All but one miRNA sequence fall within hairpins comprised of base pairs with average z-score <-2 (Fig. 1A; ebv-BART-5 miRNAs occurred in a hairpin comprised of z-score <-1 base pairs). Features of the `ScanFold-Fold` model secondary structures for the EBV miRNAs are consistent with what is known about active precursor (pre)-miRNA hairpins. For example, the ebv-BART3* and ebv-BART3 sequences are annealed to each other, are offset by two nucleotides and contain internal bulge loops (Fig. 1B). A similarly sized hairpin was detected elsewhere in the EBV genome, which corresponds to the EBV viral small nucleolar (v-sno)RNA1 (Fig. 2) (*Hutzinger et al., 2009*). All model pairs for the v-snoRNA are correctly predicted (consistent with C/D box snoRNAs); however, the average z-scores, while negative, were not lower than −1 (Fig. 2A). Thirteen additional base pairs are predicted beyond the annotated v-snoRNA region (orange pairs in Fig. 2B).

`ScanFold-Fold` was able to detect the terminal hairpin loops of very large hairpin structures found in ncRNAs from two regions. The first hairpin (Fig. S1) occurs in the W repeat region, which generates the stable intronic sequence (sis)RNA-2. This sequence occurs 5-8X in circulating viral strains and the transcribed sisRNAs accumulate to high abundance in infected cells (*Moss & Steitz, 2013*) and appear to be important to EBV-induced transformation of human B cells (*Bridges et al., 2019*; *Szymula et al., 2018*). The second region is the latent origin of replication (oriP), which is transcribed bidirectionally in lytically reactivated cells. In both oriP transcripts, tandem repeat sequences anneal to each other to form very long hairpins, which interact with human ADAR (adenosine

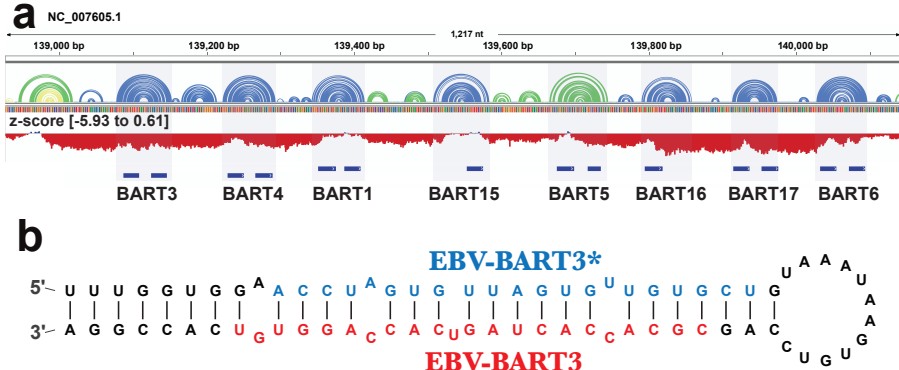

**Figure 1   Results for the EBV-BART miRNA cluster.** (A) IGV visualization of results. EBV-1 genome co-ordinates are followed by `ScanFold-Fold` predicted base pairs represented as arcs (colored blue and green for bp with average z-score < −2 and −1, respectively, and yellow for bp with negative average z-scores > −1), and the genome sequence (A, C, G, and T are in green, blue, orange and red, respectively). Below this is a bar-graph showing the window z-score values predicted by `ScanFold-Scan` (each bar sits at the 1st nt of the window and is colored red or blue for negative/positive values, respectively; the range of values are indicated in brackets). A cartoon of the gene structure of EBV-1 is shown in blue with the location of BART miRNAs annotated. (B) At the bottom is the `ScanFold-Fold` secondary structure model for the EBV-BART3 pre-miRNA hairpin. Mature sequences are annotated in blue and red.

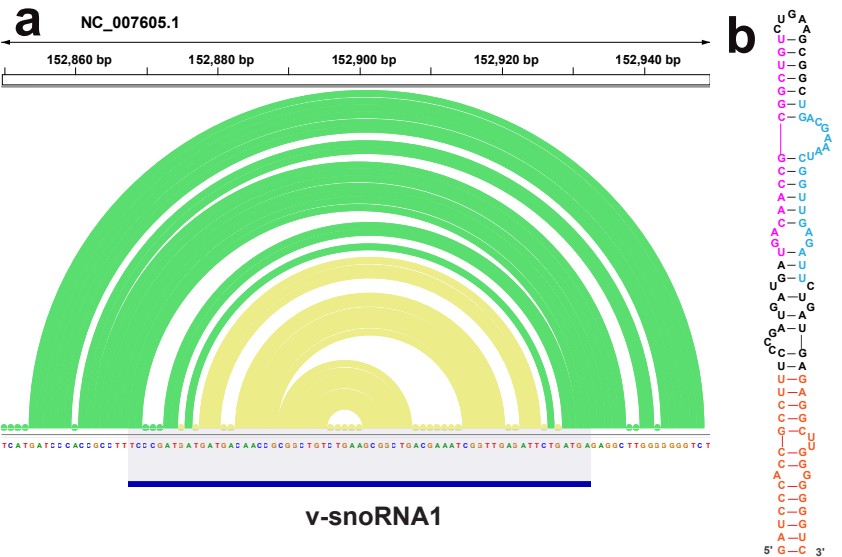

**Figure 2   Results for the v-snoRNA1.** (A) IGV visualization of results. EBV-1 genome coordinates are followed by `ScanFold-Fold` predicted base pairs represented as arcs (colored green for bp with average z-score < −1 and yellow for bp with negative average z-scores > −2), the genome sequence (A, C, G, and T are in green, blue, orange and red, respectively). (B) The `ScanFold-Fold` secondary structure model for v-snoRNA1 where additional sequences, beyond the annotated v-snoRNA1 sequence is colored orange. Proposed D' and D interaction sites with host rRNA sequences are indicated in pink and blue, respectively.

deaminase acting on RNA), are hyper-edited, localized to the nucleus, interact with paraspeckle assembly factors, and play likely roles in immune modulation (*Cao et al., 2015a*). ScanFold-Scan finds the terminal hairpins of each oriP transcript in windows with less than −5 z-scores, however, the final model reported by the ScanFold-Fold algorithm predicts multiple local inter-repeat interactions (Fig. S2). In both the sisRNA-2 and oriP transcripts, long range base pairs in each hairpin cannot be predicted, as they span distances greater than the window size used (150 nt here). This highlights an important limitation of ScanFold and, indeed, all scanning window approaches.

ScanFold-Fold was able to successfully predict most of the known structure of the EBV encoded small RNA (EBER)1 (Fig. 3), a hyper-abundant viral ncRNA with implications to oncogenesis (*Iwakiri, 2016*; *Moss et al., 2014*). Similar to the long hairpins discussed above, EBER1 long-range base pairs were not predicted, however, 61% of the pairs not spanning more than 150 nt were correctly predicted; the majority (14/15) with average z-scores <-2. In EBER2, however, no model base pairs were predicted (Fig. 4) and, indeed, few base pairs with low z-scores were predicted. The 21 base pairs with average z-scores <-1 do not correspond to any known EBER2 structures. This is due to the poorer predicted metrics for windows spanning the annotated EBER2 region; indeed, windows spanning EBER2 had positive z-scores, indicating a sequence that is potentially ordered to be loosely structured (*Andrews, Baber & Moss, 2019*). Interestingly, the regions of EBER2 that fall within positive z-score regions correspond to an interaction site that binds to nascent transcripts from the terminal repeat (TR) region of the EBV genome. This interaction facilitates the recruitment of the human PAX5 regulatory protein to the TR genomic region to promote lytic replication (*Lee et al., 2015*). Interestingly, ScanFold predictions in the TR region find that the EBER2 interaction sites are *also* spanned by positive z-score windows and ScanFold-Fold predictions suggest that these regions are ordered to be unstructured/accessible.

## New structural motifs in EBV

In addition to finding known EBV structures, ScanFold predicts many additional motifs with putative functions. We focused our attention on one region which was particularly rich in base pairs predicted to have average z-scores that were exceptionally low ($< −2$). This region approximately spans EBV bp 48,800 to 50,200 and overlaps several important viral genes (Fig. 5). The first cluster of predicted motifs overlaps three lytic genes BFRF1, BFRF2, and BFRF3, which play essential roles in nuclear egress (*Farina et al., 2005*) (the first step in virion release into infected cells), regulation of late viral gene expression (*Aubry et al., 2014*), and assist in the assembly of infections particles (*Henson et al., 2009*), respectively. Each gene overlaps each other, where BFRF2 and BFRF3 both overlap the 3′ UTR of BFRF1 and BFRF3 overlaps the 3′ UTR of BFRF2. Thus, functional structural motifs can be playing multiple roles in the post-transcriptional control of BFRF1-3. We focused our attention, however, on potential roles in the 3′ UTR of BFRF1. A 606 nt fragment sequence (bp 48,856 −49,461), corresponding to the motifs predicted with average z-scores less than -1 in the BFRF1 3′ UTR were further analyzed.

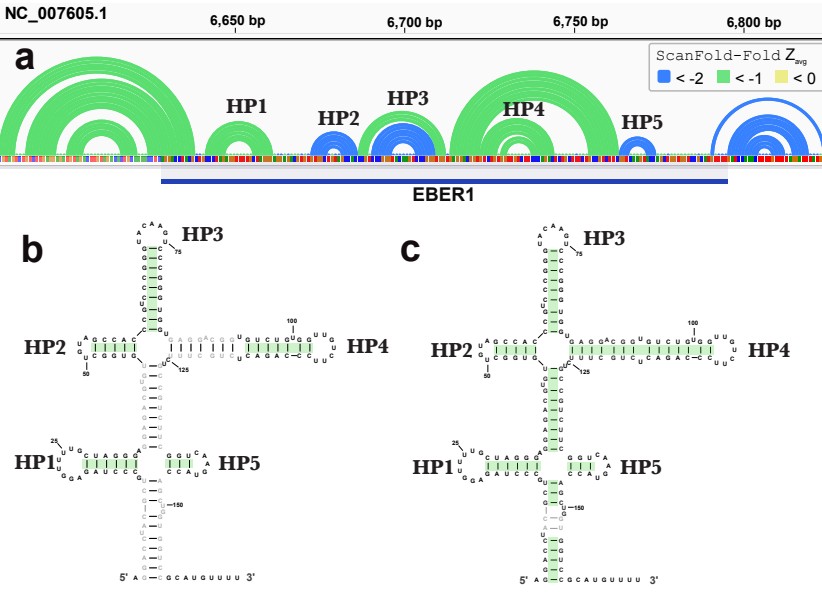

**Figure 3 Results for EBER1.** (A) IGV visualization of results. At the top of IGV, EBV-1 genome coordinates are shown, followed by ScanFold-Fold predicted base pairs represented as arcs and the location of EBER1 is indicated by the blue cartoon. Base pairs in the reference EBER1 structure correctly predicted by ScanFold-Fold are highlighted in green on the secondary structure in (B). (C) indicates the correctly predicted pairs after refolding the EBER1 sequence using ScanFold-Fold bp with average z-scores < −2 as constraints.

All base pairs predicted by ScanFold-Fold to have average z-scores <-1 were constrained to be paired to each other and the remaining sequence was refolded using the RNAfold algorithm (the same algorithm used in ScanFold-Scan to predict windows (*Lorenz et al., 2011*), which predicts the global MFE structure. This fills in base pairs that are "missing" in the ScanFold-Fold consensus (Fig. 6A), allows for potential long-range interactions to occur and allows for base pairs that do not contribute to the low z-score of this region to form. The resulting global model of this region is shown in Fig. 6B. The overall fold can be divided into eight motifs. In most cases a motif was simply the ScanFold-Fold predicted structure. In Motif (M)2 and M6, several base pairs were added to extend helices (Fig. 6B). M4, on the other hand, was predicted to form a multibranch loop structure, which encompasses three ScanFold-Fold predicted hairpins: M4.1, M4.3 and M4.5. To test the resulting model, a phylogenetic analysis was undertaken. The EBV-1 sequence was queried against the nt database using BLASTn (*Altschul et al., 1990*), which identified 100 putative homologs, all from other EBV strains. Model base pairs were compared to an alignment of EBV-1 and these 100 homologous sequences (Data S1). Canonical base pairing was conserved 99.1% and, when mutations occurred, they generally preserved structure. Of the 202 base pairs predicted in this region, 35 showed evidence of consistent mutation—a single point mutation which preserves base pairing (Fig. 6B). Only two base pairs showed evidence of compensatory mutation—a double point mutation which

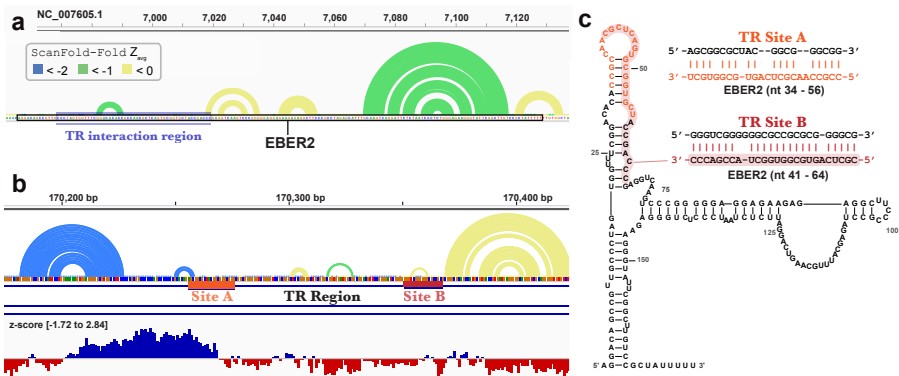

**Figure 4 Results for EBER2 and its interaction sites on the terminal repeat RNA sequence.** (A) EBV-1 genome coordinates are followed by ScanFold-Fold predicted base pairs represented as arcs (colored blue and green for bp with average z-score $< -2$ and $-1$, respectively, and yellow for bp with negative average z-scores $> -1$), the genome sequence (A, C, G, and T are in green, blue, orange and red, respectively). The EBER2 sequence is outlined with a black box, this includes an annotation of the terminal repeat (TR) interacting nucleotides in EBER2. (B) ScanFold-Fold predicted base pairs for the TR region, the corresponding binding region to EBER2, represented in panel A, with the addition of annotations for the two EBER2 interaction regions (sites A and B) and a bar-graph showing the window z-score values predicted by ScanFold-Scan (each bar sits at the 1 st nt of the window and is colored red or blue for negative/positive values, respectively; the range of values are indicated in brackets). (C) Secondary structure model of EBER2 with the TR interacting nucleotides annotated in orange and red for Site A and B, respectively. Model duplexes for each interaction are shown to the right.

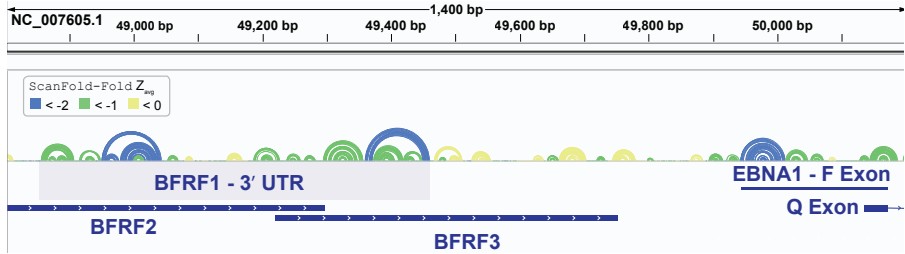

**Figure 5 Results for the EBV-1 genome region partially encoding BFRF1-3 and part of EBNA1.** EBV-1 genome coordinates are followed by ScanFold-Fold predicted base pairs represented as colored arcs and the location of each gene is indicated by the blue cartoon. The light blue highlighted region is the region used for subsequent experimental analysis.

preserves base pairing. One putative compensatory mutation occurs in the basal stem of M4.5 and the other in M7 (Fig. 6B).

A second cluster of predicted motifs overlaps the F/Q exons of EBNA1 (Fig. 5). EBNA1 is a key viral gene involved in the replication and partitioning of the EBV genome (*Frappier, 2012*). During latency program I, EBNA1 is the sole expressed protein and transcription begins at the Q promoter (Qp; Fig. 7A). This results in the production of a short Q exon, which is spliced upstream of the U leader exon in the EBNA1 5′ UTR, which was previously found to contain a structured internal ribosomal entry site (IRES) that stimulates non-canonical (cap-independent) translation (*Isaksson, Berggren & Ricksten, 2003b*). During

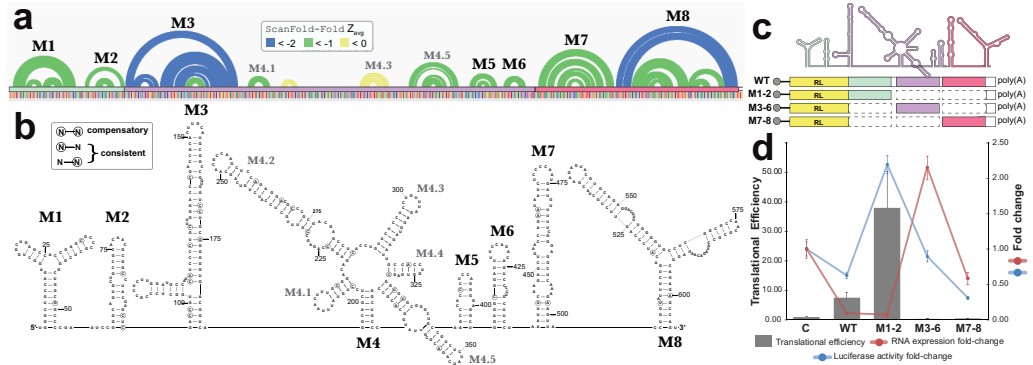

**Figure 6** **Results for the region partially overlapping BFRF1-3.** (A) The top of this panel shows ScanFold-Fold predicted base pairs represented as colored arcs. (B) Secondary structure models predicted after energy minimization using ScanFold-Fold base pairs (average z-score < −1) as constraints. Eight motifs (M1-8) were defined (labeled in both the (A) and (B). Nucleotides with mutations preserving the shown base pair are circled on the structure models. (C) Cartoon showing the locations of three regions (colored green, purple, and red) that were added to downstream of *Renilla* luciferase (RL; diagrams of constructs are shown with RL in yellow and the added fragments of EBV-1 in colors corresponding to the cartoon). (D) Results of luciferase assays for each construct; the translational efficiency and luciferase mRNA and protein fold-change (vs. empty vector control (C)) are plotted for C, wild type (WT) and three fragments of the structured region. The translational efficiencies reported here were calculated by dividing the relative protein abundance of RL (RRR values) by mRNA levels of RL ($2^{-\Delta\Delta CT}$ values); see methods for detail and Table S2 for raw values.

lytic reactivation transcription from the F̲ promoter (Fp) leads to the incorporation of the longer F exon into the 5′ UTR of EBNA1 (includes the Q exon sequence; Fig. 7A). To see if the ScanFold-Fold predicted structural motifs would persist in the context of this longer 5′ UTR, the entire sequence (plus the start codon), were analyzed using the ScanFold pipeline. Three hairpins were predicted (HP1-3; Fig. 7B) which recapitulate those predicted in the EBV genome scan (Fig. 5). The most significant base pairs (average z-scores < −2) all occur within HP1, which begins at the first nucleotide transcribed from Fp.

In the context of the 5′ UTR, the downstream F/Q exon nt are predicted to form a novel hairpin, HP4. No significant base pairing was predicted between the F/Q and U leader exons. Interestingly, no base pairs were predicted in the IRES-containing U leader exon with average z-scores <-1. The negative z-score (< 0 but > −1) base pairs predicted here do not correspond to any found in the EBNA1 IRES model structure (*Moss et al., 2014*). After constraining ScanFold-Fold predicted base pairs (<-1) and re-folding the 5′ UTR, however, four additional hairpins (HP5-8) were predicted (Fig. 7C). HP7 and HP8 are found in the current consensus model of the EBNA1 IRES (Rfam ID# RF00448) in the Rfam database (*Burge et al., 2013*; *Daub et al., 2015*; *Gardner et al., 2009*; *Griffiths-Jones et al., 2003*; *Griffiths-Jones et al., 2005*; *Kalvari et al., 2018a*; *Kalvari et al., 2018b*; *Nawrocki et al., 2015*). HP6 is a novel predicted configuration of the IRES nucleotides and HP5 is a novel predicted hairpin upstream of the IRES, which partially overlaps (incorporating two

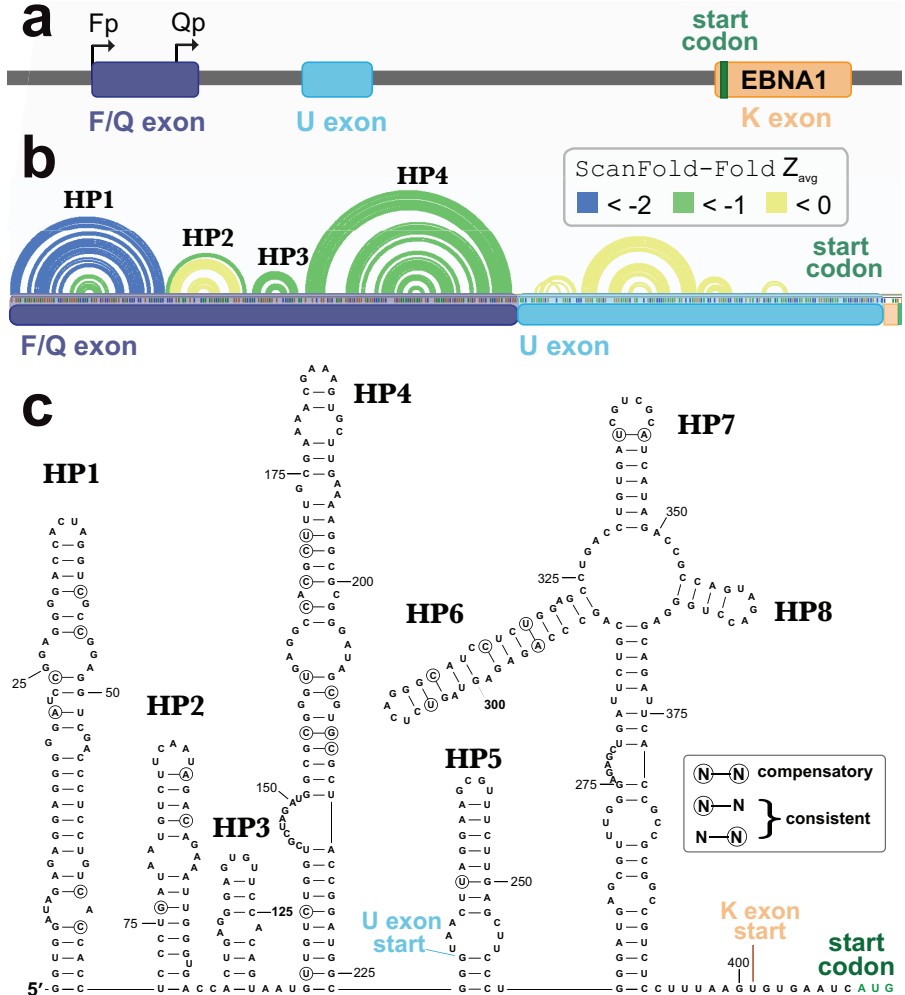

**Figure 7** **Results for the EBNA1 5′ UTR.** (A) Cartoon showing the EBV-1 genome region with the Fp (used in lytic replication) and Qp (used in latency type I and II) promoters annotated alongside the F/Q, U, and K exons (with start codon colored green). (B) ScanFold-Fold predicted base pairs represented as colored arcs above the sequence of the EBNA1 5′ UTR formed after transcription from Fp. Spliced exon sequences are highlighted and labeled below the arc diagram structure (with highlights colored as in the cartoon). (C) Secondary structure model of the EBNA1 5′ UTR based on energy minimization using ScanFold-Fold base pairs with average z-scores < −1 as constraints. The locations of exon starts and the start codon are annotated; as well, nt that show evidence of undergoing structure-preserving mutations are circled.

nucleotides of) the F/Q exon (Fig. 7C). HP6-8 are predicted to occur within a multibranch loop structure that sits above an extensive basal stem structure.

To check for potential conservation of the model structure, as well as identify putative consistent and compensatory changes to support model base pairs, a phylogenetic comparison was performed using an alignment of 49 EBV sequences (Data S2). Overall conservation of canonical base pairing was 98.0%. There were 22 putative consistent mutations and 3 compensatory mutations identified (Fig. 7C). These putative

compensatory mutations are found in HP4, HP6 and HP7. No consistent or compensatory mutations were identified within the long basal stem under HP6-8.

## Experimental analysis of a novel structured region

To test potential functions of the motifs overlapping BFRF1, BFRF2, and BFRF3, four constructs were generated that comprise the entire WT structural region and three fragments (containing M1-2, M3-6, and M7-8; Fig. 6C), which were placed downstream (within the 3′ UTR) of Renilla Luciferase (RL) and expressed in human cell lines. Two values were measured for each construct, as well as an empty vector control (Fig. 6D): the Luciferase activity fold-change (Luciferase activity is a measure of protein expression levels) and the translational efficiency (which normalizes protein levels using the RNA abundance measured by RT-qPCR); see Methods. Compared to the control (C), the WT construct had reduced Luciferase activity, but a ∼7-fold increase in translational efficiency. The M1-2 construct saw marked increases in Luciferase activity (∼2-fold) and translational efficiency (∼36-fold increase). M3-6 reduced the Luciferase activity to wild type levels, however translational efficiency was roughly cut in half compared to control. M7-8 reduced the Luciferase activity and translational efficiency below WT levels (by roughly 60% in both cases). These results indicate the presence of repressive elements in the region spanned by M3-8 and stimulatory elements in M1-2.

## Structural motifs beyond EBV

Beyond EBV, RNA secondary structures have been previously discovered in KSHV. A recent analysis of the KSHV PAN ncRNA (Sztuba-Solinska et al., 2017) (an RNA transcript that is important to late lytic gene expression and the release of progeny virions (Borah et al., 2011; Sun et al., 1996)) generated a global secondary structure model for PAN based on comprehensive SHAPE probing under in/ex cellulo, and in/ex virio in vitro conditions. ScanFold−Fold results for the KSHV genomic region encoding PAN recapitulated many features of the experimentally-informed global model (Fig. 8). Notable, however, is the lack of highly negative average z-score base pairs from the ScanFold−Fold predictions: indeed, the majority of base pairs were predicted with negative average z-scores that did not fall below −1. Base pairs predicted with average z-scores below −1 clustered between PAN nucleotides 137 to 327; the only motif from the previous global model with average z-score < −1 fell within this region (H8; Fig. 8).

Motif base pairs (after refolding; see Methods) are 99.4% conserved in an alignment of PAN homologs (Data S3) and the little variation observed was consistent with base pairing. 11 out of 19 motifs predicted by ScanFold appeared in the global model (Fig. 8) (Sztuba-Solinska et al., 2017) and, with small variations (e.g., see results of the PAN ENE below), recapitulated the experimentally-guided model base pairs. The nucleotides in the 8 ScanFold motifs that were not found in the global model were, in that model, generally found in longer range interactions and/or showed high SHAPE reactivity under various probing conditions (indicating single-stranded or loose structure). These results are consistent with the observation that regions with z-scores greater than -1 generally corresponded to higher SHAPE reactivities (Andrews & Moss, 2019) (potentially due to the

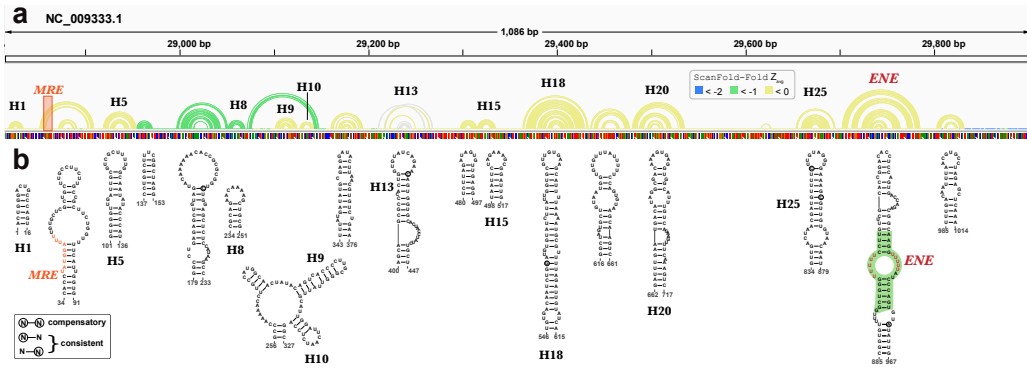

**Figure 8 Results for KSHV PAN.** (A) IGV visualization of results with KSHV genome coordinates followed by ScanFold-Fold predicted base pairs represented as arcs (colored green for bp with average z-score $< -1$, yellow for bp with negative average z-scores $> -1$, and grey for $> 0$), the genome sequence (A, C, G, and T are in green, blue, orange and red, respectively). (B) The bottom panel contains motif secondary structures predicted by ScanFold-Fold depicted using VARNA (*Darty, Denise & Ponty, 2009*). The MRE is represented as an orange highlight on the arc diagrams and as orange bps in the secondary structures. The nucleotides from the ENE motif which have been crystalized (*Mitton-Fry et al., 2010*) are highlighted in green.

lack of a predominant secondary structure and/or the presence of primary sequence motifs whose activity is facilitated by being unstructured). For example, the motif predicted for nucleotides 34 to 91 contains the Mta responsive element (MRE; a protein binding site that stabilizes PAN (*Massimelli et al., 2011*; *Tunnicliffe et al., 2019*) partially occluded within a weak stem (comprised of GU and AU base pairs; Fig. 8). In the global model the MRE site is highly reactive to SHAPE probing (under ex virio conditions) and the remaining motif bases are contained within long-range helices (H2 and H3 in Fig. 1 of (*Sztuba-Solinska et al., 2017*). The motif spanning nucleotides 343 to 376, however, is modeled as a small alternative hairpin (H12 in Fig. 1 of (*Sztuba-Solinska et al., 2017*) in a looped out region with moderate SHAPE reactivity (under ex virio conditions).

The best characterized (in structure and function) motif is the ENE (expression and nuclear retention element) found in the 3′ end of the KSHV PAN ncRNA (*Conrad et al., 2006*). This element was found to increase the stability/lifetime of intronless transcripts, such as the PAN RNA, via the sequestration of the PAN poly(A) tail in an extensive triple-helix structure with bulged uracil nucleotides in the ENE (*Mitton-Fry et al., 2010*). In the analysis of the KSHV genome, ScanFold-Fold was able deduce the location of the ENE and predict a secondary structure in agreement (Fig. 8) with the crystal structure of the PAN ENE(*Mitton-Fry et al., 2010*) (crystalized nucleotides highlighted in green) and the SHAPE directed model (of the cytoplasmic and *ex virio* RNA; Figs. S4 and S7 of *Sztuba-Solinska et al. (2017)*).

Beyond the KSHV PAN there are many base pairs predicted with higher significance: there are 39906 base pairs that have average z-scores $< -1$ and 9446 $< -2$ predicted in the KSHV genome. The values for KSHV are on par with predictions for other herpesviruses (Table 1), indicating significant fractions of each genome likely encode functional RNA

secondary structures (all results available on the RNAstructuromeDB (*Andrews, Baber & Moss, 2017*) and in Supplemental Data set (https://zenodo.org/record/3964325).

## DISCUSSION

Innovations present in the `ScanFold` approach made this current report possible. The initial `ScanFold-Scan` analysis generates metrics of local RNA folding that allow for genome-wide comparisons between human herpesviruses. A key finding of this study is that EBV (types 1 and 2) both have evidence of containing more sequences that are ordered to form (likely functional) RNA structures than any other herpesviruses: e.g., the average z-score for EBV is ~2X that of the next lowest (KSHV; Table 1). It should be noted, however, that even for the genome with the least negative average z-score, HHV-6B ($-0.09$; Table 1), significant percentages of the genome are spanned by low z-score windows. These give rise to many predicted base pairs with likely functionality: e.g., for HHV-6B 5.8% and 0.8% of the 607,730 base pairs predicted by `ScanFold-Fold` have average z-scores $< -1$ (28,525 bp) and $< -2$ (4,780 bp), respectively. Thus, although RNA secondary structure is predicted to play greater roles in EBV, all human herpesviruses are likely to utilize regulatory RNA structure to a larger extent than previously appreciated. The results of these analyses are made available on the RNAstructuromeDB (*Andrews, Baber & Moss, 2017*) to facilitate their usage by a wide array of researchers interested in human herpesviruses. Here, results can be browsed alongside genomic annotations (which have been loaded from NCBI for each genome) or any JBrowse compatible file (*Buels et al., 2016*) a user has available. As an example, for EBV-1 we include additional tracks alongside `ScanFold` results to help identify regions of interest; the McIntosh lab has generated RNA sequencing data for the EBV-1 genome as it transitions from a latent to a lytically active state (*Frey et al., 2020*). The sequencing data from this study can now be seen as a coverage track and allows users to quickly assess whether regions highlighted by `ScanFold` fall within actively transcribed regions.

In addition to facilitating global comparisons, the `ScanFold` approach defines specific motifs of likely function. All predicted motifs containing at least one base pair with an average z-score <-2 have been modeled, annotated, and compiled into individual PDF files for easy viewing (Supplemental Data set - https://zenodo.org/record/3964325). In addition to presenting the thermodynamic properties for each motif, the NCBI gene features overlapping the motif have been listed. In EBV-1, `ScanFold` was able to identify the locations of and recapitulate elements of known functional structured RNAs (Figs. 1–3 and Figs. S1–S2). For example, although `ScanFold` is not explicitly a miRNA discovery program, it does a strikingly good job of deducing pre-miRNA hairpins. All known EBV miRNAs are contained within hairpins formed by base pairs with z-averages $< -1$ and the majority have base pairs $< -2$ (RNAstructuromeDB and Supplemental Data set - https://zenodo.org/record/3964325: e.g., the BART miRNA cluster shown in Fig. 1. The exceptionally low average z-scores of pre-miRNA base pairs indicates a particular bias in their sequence order, highlighting the strict structural requirements of miRNA maturation (*Lee et al., 2002*; *Zeng & Cullen, 2003*).

ScanFold results may also provide information on the processing of another EBV ncRNA, v-snoRNA1. In addition to predicting a C/D box snoRNA-like hairpin in the annotated v-snoRNA1 region, additional base pairs are predicted that lengthen the hairpin (Fig. 2). Interestingly, these additional predicted base pairs have lower average z-scores than those of the core v-snoRNA1 structure. The extended hairpin predicted here may be a precursor structure, required to mature the v-snoRNA1 (*Hutzinger et al., 2009*; *Matera, Terns & Terns, 2007*). It is also interesting that, while being negative, the majority of core base pairs are not predicted to have average z-scores below -1. This may be to facilitate intermolecular RNA-RNA interactions with target sites on the host RNAs: e.g., utilizing the D' and D sites (*Hutzinger et al., 2009*) annotated on Fig. 2.

Similar results were found in the EBER2 and TR regions of EBV, which form biologically essential intermolecular RNA-RNA interactions with each other (Fig. 4) (*Lee et al., 2015*). But for several isolated base pairs, the EBER2 TR interaction region is predicted to have unusually stable base pairs (Fig. 4A) as are the two interaction sites in the TR RNA (Fig. 4B), which occur within a region of highly positive predicted z-score—indicating they may be ordered to be unusually unstable (presumably to facilitate intermolecular base pairing). None of the EBER2 reference structure (Rfam ID# RF02712) base pairs are predicted by ScanFold-Fold (Fig. 4C); emphasizing its loose secondary structure, which is also supported by previous experimental analyses (*Lee et al., 2015*).

In contrast to EBER2, EBER1 is predicted to contain base pairs with average z-scores $< -2$, indicating a high degree of ordered structure, which is corroborated by experimental analyses (*Glickman, Howe & Steitz, 1988*; *Lee et al., 2015*). These low z-score base pairs form three hairpins in the reference structure (Rfam ID# RF01789), with the two remaining reference structure hairpins being predicted by base pairs with average z-score <-1 (Fig. 3). The remaining 46% of base pairs in the reference structure that were not predicted by ScanFold-Fold highlight an important limitation of ScanFold: it cannot predict longer range base pairs (due to the limitation of the window size used). To predict the reference structure, however, ScanFold-Fold results can aid modeling by providing base pairs to use as constraints (e.g., Fig. 3C). *A priori* knowledge of the final transcript sequence is, however, essential here. This highlights another important limitation of whole genome scans: defining the boundaries of a structured domain or ncRNA. In EBER1, for example, 10 base pairs with average z-scores <-2 are predicted between EBER1 and upstream EBV sequences; as is a single base pair <-1 predicted to occur downstream (Fig. 3A). The EBER1 sequence can be co-transcribed as part of intronic sequences for latent membrane protein (LMP)2 and these structures between EBER2 and flanking sequence could play some role in splicing; however, additional work is needed to test this speculative hypothesis.

Beyond previously described structures in EBV, ScanFold uncovered many novel motifs; we focused on structures predicted in a particular "hot spot" with base pairs having average z-scores <-2 (indicating highly ordered sequences/structures; Fig. 5). The motifs overlapping BFRF1-3 do so primarily in 3′ UTR sequences (exclusively in BFRF1 and partially in BFRF2), thus their hypothesized roles in regulation of gene expression; as 3′ UTRs are particularly rich areas for post-transcriptional control elements (*Mayr, 2017*). Indeed, we found that inclusion of motifs M1 and M2 into a reporter construct

stimulated expression of luciferase, while motifs M3 to M6 and M7 to M8 both suppressed expression (Fig. 6B). Many factors can be playing roles in the observed effects of these motifs. For example, multiple host regulatory proteins are predicted to target M1-8 (Data S5) as are both host and viral miRNAs (Table S1). Motif structure can play many roles here: by forming specific recognizable motifs; occluding or presenting primary sequence motifs, or by altering the relative positioning of regulatory motifs. Much additional work is required to validate predicted interactions in this region and to define the exact roles of the conserved and unusually stable secondary structures discovered here: e.g., introducing minimal mutations to modulate predicted RNA secondary structures can help determine to what extent RNA structure plays a role in the observed results.

Similarly, we identified four highly stable/conserved hairpins in the F/Q exon of EBNA1, which are, essentially, appended as a single domain to the 5′ UTR of this critical viral gene during lytic replication. An interesting observation was that these novel elements had base pairs that were predicted with average z-scores that were much lower than those of the known structures in the downstream U leader exon (*Isaksson, Berggren & Ricksten, 2003a*) (Fig. 7). Indeed, the EBNA1 IRES base pairs could only be predicted after refolding the 5′ UTR with upstream sequences constrained based on the `ScanFold-Fold` predicted base pairs. Phylogenic support is found for all model structures, indicating that all have likely importance; however, the IRES shows much less evidence of unusual stability. Perhaps, as in the previously discussed examples, some flexibility is required in the IRES for its function. What potential roles then, can be proposed for the rigidly defined structures in the F/Q exon? One interesting possibility is that structure here is inhibiting cap-dependent translation (e.g., by sequestering the 5′ end of the mRNA in the exceptionally stable HP1; Fig. 7), potentially driving the use of the EBNA1 IRES in non-canonical translation. This hypothesis is supported by data from a previous study (*Isaksson, Berggren & Ricksten, 2003b*), which showed that inclusion of this structured upstream sequence reduced relative luciferase activities of all tested constructs. Furthermore, when the IRES domain was not present, constructs containing the upstream structures had the lowest relative luciferase observed levels.

## CONCLUSION

A significant outcome of this study was the prediction of extensive functional RNA structures throughout other human herpesviruses. It is likely that, similar to EBV, these motifs are playing important roles in the regulation of herpesvirus biology, infection and disease. Additional studies of these motifs will provide many insights, which are facilitated by making all results available on the RNAstructuromeDB. Similarly, a web server is available (*Andrews, Baber & Moss, 2019*; *Andrews & Moss, 2019*) for running `ScanFold` calculations and for aiding in mutational design of constructs used in experimental assays (e.g., using RNA2DMut *Moss, 2018b*).

## ACKNOWLEDGEMENTS

Thanks to Nuwanthika Kumarasinghe for her assistance.

### Funding

This research was supported by NIH/NIGMS grants R00GM112877 and R01GM133810, as well as by startup funds from the Roy J. Carver Charitable Trust. The funders had no role in study design, data collection and analysis, decision to publish, or preparation of the manuscript.

### Grant Disclosures

The following grant information was disclosed by the authors:
NIH/NIGMS: R00GM112877, R01GM133810.

### Competing Interests

The authors declare there are no competing interests.

### Author Contributions

- Ryan J. Andrews and Collin A. O'Leary performed the experiments, analyzed the data, prepared figures and/or tables, authored or reviewed drafts of the paper, and approved the final draft.
- Walter N. Moss conceived and designed the experiments, performed the experiments, analyzed the data, prepared figures and/or tables, authored or reviewed drafts of the paper, and approved the final draft.

### Data Availability

ScanFold output for all genomes are available at Zenodo: Ryan Andrews, Collin O'Leary, & Walter Moss. (2020). Data S1 [Data set]. Zenodo. http://doi.org/10.5281/zenodo.3964325.

The output is also available from the RNAStructuromeDB at https://structurome.bb.iastate.edu/herpesvirus.

### Supplemental Information

Supplemental information for this article can be found online at http://dx.doi.org/10.7717/peerj.9882#supplemental-information.

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
