# Peer review of "A survey of RNA secondary structural propensity encoded within human herpesvirus genomes: global comparisons and local motifs"

_PeerJ, doi:10.7717/peerj.9882_

## Round 0.1 · original submission · Major Revisions

Our apologies for how long it took to get this ms reviewed.
In any case, two reviews were finally secured from knowledgeable investigators. Please respond point-by-point.
We look forward to receiving a revision!

·

Basic reporting

It is well established that virus genomes can harbor vast amounts of sometimes overlapping information: binding sites for transcription regulators, ORFs, polyA signals, and more. Virally-encoded RNAs can do much more than direct translation of proteins, including microRNAs and other forms of non-coding RNA. This submission relates to tools the authors have developed for genome-wide discovery of RNA structures of potential biological significance.

The authors applied their algorithm to all nine human herpesviruses. Most of the detailed descriptions relate to EBV, with one example from KSHV. The analyses for all nine viruses are available for inspection and further analysis at a publicly-accessible website hosted by the Moss lab.

It seems clear that much of value can be learned from study of non-coding functions of virally-encoded RNA. This work constitutes a significant advancement in development of tools for identifying candidates for experimental evaluation. Making a database of a set of comprehensive analyses spanning all nine human herpesviruses greatly lowers the threshold for others to engage in this line of work.

As detailed below, there are numerous issues related to details of how figures were prepared that get in the way of understanding the results.

Experimental design

I had no major issues with the experimental design.

It is good that the authors ran their analysis on genomes of all nine human herpesviruses.

A brief summary of interesting features from each virus would increase the likelihood of the information being used.

Validity of the findings

The new tools were able to independently predict previously-identified bioactive ncRNAs of EBV, plus additional EBV species that seem worthy of experimental investigation. One of the new predictions was experimentally validated.

Additional comments

1. While the work seems interesting, more needs to be done to make the work accessible and useful to people who might be able to make use of it.
a. Every type of graphical element in figures needs to be explained, and repeated in other such figures, so the reader does not need to spend a day chasing down explanatory factoids needed to understand what was done.
b. The purpose of every Supplementary table needs to be explained.
c. Every column in Supplementary tables needs to be explained.
d. Each of the Word files that contain collections of FASTA-formatted sequences needs an understanding-enabling explanation of what was included and why.
e. In the figures, sequence coordinates need to be legible without magnification; if the numbers come out of some software that uses characters too small to be read, they need to be replaced with legible numerals in the version intended for publication.

2. Fig. 4a includes several horizontal lines above “EBER2”. What do they mean? How is the reader supposed to decipher intent vs. irrelevant graphical residue. Similar questions about Figs. 5 and 8.

3. In Fig. 7, the colors used in the upper and second panel are described as being the same for the representations of the F/Q and U exons, but the colors don’t actually match, which places an unnecessary burden on the reader to decipher the intended meaning. Few readers will persist.

4. Table 1 mentions large numbers of “total bp” for each virus. It is not clear how the numbers connect to the statement in the legend that “Total base pairs gives the number of stable pairs predicted in each genome”. For HCMV, how do you go from a 236 kb genome to 1.13 million total bp? Similar question for the others.

5. The Supplemental Fig. 2 legend mentions red and blue bars for Z-values, but the entire plot is blue. The graph says the scale ranges from -8.380 to 3.87, but that does not make sense relative to the graph that is shown (small y-values).

6. In figures and at the website, I would like to see the folding information presented in the context of major well-understood genomic features, such as ORFs, polyA signals, etc. Including two or three tracks for display of such information as available in Genbank features tables would be helpful to many users.

7. It is good that the authors ran their analysis on genomes of all nine human herpesviruses. A brief summary of interesting features from each would increase the likelihood of the information being used.

8. Is information available from RNAseq data that might provide additional validity for the presence of some of the predicted ncRNAs?

Suggested edits:
Abstract: EBV is not the only herpesvirus with very high prevalence. VZV, HHV-6B, and HHV-7 are in the same ballpark as described in the abstract for EBV. A friend once screened >1500 specimens for antibodies against HHV-6 and found only one clear negative.

Abstract: I am not sure what is meant by “transcriptome-wide”. It seems like your analysis is based on examination of genome sequences, not on products of RNAseq.

Line 63 and 167: In general, “herpesvirus” is better as one word. Being consistent in usage is good, too.

Line 70: Order Herpesvirales, Family Herpesviridae, Subfamily Gammaherpesvirus, Genus Lymphocryptovirus. Italicize the Latinized taxon names.

Line 74. “herpes simplex viruses 1 and 2”

Line 104. It was not clear how the closely related lymphocryptovirus was used.

Line 206. Something is missing after “in”.

Line 238. “Whole cell RNA”

Reviewer 2 ·

Basic reporting

This is a clearly written new comprehensive study of folding predictions for all member of herpesviruses known to infect humans. The background and introduction set the stage perfectly for the study although the sentence beginning at line 102 (“Here the ncRNA…”) seems to fit better within the last paragraph of the introduction (the text that follows the current positioning refers to previous studies, whereas this sentence implies a reference to the current study). The figures are all of high quality and are state of the art for presenting this type of data. Further, they clearly illustrate the points being addressed. All aspects of experimental design are well laid out and derived from previous groundwork.

Experimental design

All experiments are well designed that are well laid out and presented in a fashion that will allow for reproducibility

Validity of the findings

The findings are presented in an appropriate scientific manner and outline the strengths and limitations for the results.

Additional comments

Overall, this is a nice comprehensive study of folding predictions across the human herpesvirus clades. It is refreshing that a validation experiment is provided which suggests functional relevance of sequences within the BFRF1 3’ UTR that are predicted to form hairpins. The reporter system certainly appears to demonstrate different impacts on RNA stability and translation efficiency although it would have been nice if the authors had included mutational studies to formally demonstrate that the predicted structures (rather than the sequences themselves) are functional.

Overall, this is a well presented comprehensive analysis of predicted human herpesvirus structures and lays important groundwork for future functional studies.

---

## Round 0.2 · accepted · Accept

The thoughtful edits and responses were appreciated.

The original review took an unusually long time so I read the paper myself and was satisfied that the authors had done a great job. Thus, I decided to write directly to the two reviewers, including the revised ms PDF and response to ask them both whether they wanted to see this ms again or were satisfied, indicating that it met my expectations as an editor. Both were happy for this revision to be accepted.